# Radiographic, Biomechanical and Histological Characterization of Femoral Fracture Healing in Aged CD-1 Mice

**DOI:** 10.3390/bioengineering10020275

**Published:** 2023-02-20

**Authors:** Maximilian M. Menger, Ruben Manuschewski, Sabrina Ehnert, Mika F. Rollmann, Tanja C. Maisenbacher, Anne L. Tobias, Michael D. Menger, Matthias W. Laschke, Tina Histing

**Affiliations:** 1Department of Trauma and Reconstructive Surgery, Eberhard Karls University Tübingen, BG Trauma Center Tübingen, 72076 Tübingen, Germany; 2Institute for Clinical & Experimental Surgery, Saarland University, 66421 Homburg, Germany; 3Siegfried Weller Research Institute, Department of Trauma and Reconstructive Surgery, Eberhard Karls University of Tübingen, BG Trauma Center Tübingen, 72076 Tübingen, Germany

**Keywords:** mice, bone healing, bending stiffness, senescence, aging

## Abstract

With a gradually increasing elderly population, the treatment of geriatric patients represents a major challenge for trauma and reconstructive surgery. Although, it is well established that aging affects bone metabolism, it is still controversial if aging impairs bone healing. Accordingly, we investigated fracture healing in young adult (3–4 months) and aged (16–18 months) CD-1 mice using a stable closed femoral fracture model. Bone healing was analyzed by radiographic, biomechanical and histological analysis at 1, 2, 3, 4 and 5 weeks after fracture. Our results demonstrated an increased callus diameter to femoral diameter ratio in aged animals at later time points of fracture healing when compared to young adult mice. Moreover, our biomechanical analysis revealed a significantly decreased bending stiffness at 3 and 4 weeks after fracture in aged animals. In contrast, at 5 weeks after fracture, the analysis showed no significant difference in bending stiffness between the two study groups. Additional histological analysis showed a delayed endochondral ossification in aged animals as well as a higher amounts of fibrous tissue at early healing time points. These findings indicate a delayed process of callus remodeling in aged CD-1 mice, resulting in a delayed fracture healing when compared to young adult animals. However, the overall healing capacity of the fractured femora was not affected by aging.

## 1. Introduction

There is a steadily growing elderly population, making the treatment of geriatric patients a major challenge for trauma and reconstructive surgery [1]. Aged individuals are not only associated with a higher risk for fractures but also suffer from an increased morbidity and mortality [2,3]. Moreover, it is well known that aging affects the mechanical and biological properties that determine the process of bone regeneration by degenerative comorbidities such as peripheral arterial occlusive disease, diabetes mellitus and osteoporosis. In fact, there is increasing evidence that the processes of angiogenesis and inflammation as well as bone metabolism within the skeletal system are severely affected by age-related changes [4]. However, it is still unclear if aging impairs fracture healing.

Murine animal models are of increasing interest in trauma and orthopedic research, due to low husbandry costs, gene-targeted strains and a large array of monoclonal antibodies. These antibodies allow targeting individual molecules in vivo, thereby evaluating their contribution to cell differentiation and proliferation during the process of bone regeneration [5]. Accordingly, recent developments and innovative techniques in small animal fracture fixation make murine bone healing models powerful tools in the field of preclinical research [6].

The CD-1 outbred mouse has been used in a variety of experimental studies to investigate the effects of pharmacological substances on fracture healing [7,8,9] and develop novel treatment strategies to facilitate bone regeneration [10,11]. Additionally, CD-1 mice have been successfully applied in orthopedic research, which focuses on fracture healing in aged animals [12,13]. However, to our knowledge, no study has yet compared the radiographic, biomechanical and histological healing properties of aged animals with those of young adult CD-1 mice.

Hence, in the present study, we investigated bone healing in aged and young adult CD-1 mice by X-rays, biomechanics and histology using a well standardized, stable closed femoral fracture healing model [14]. Due to age-related, detrimental effects on the mechanical and biological properties of fracture healing, we hypothesized an impaired healing capacity in aged CD-1 mice.

## 2. Materials and Methods

### 2.1. Animals

For the present study, a total of 100 male and female CD-1 mice with an age of 3–4 months (young adult, *n* = 50) and 16–18 months (aged, *n* = 50) were used. The age of 16–18 months was chosen due to previous experimental reports on age-related changes in CD-1 mice [15]. The animals were bred at the Institute for Clinical & Experimental Surgery, Saarland University, and kept at a regular light and dark cycle with free access to tap water and standard pellet food (Altromin, Lage, Germany). All animal procedures were performed according to the National Institutes of Health guidelines for the use of experimental animals and the German legislation on the protection of animals. The experiments were approved by the local governmental animal welfare committee.

### 2.2. Surgical Procedure

Mice were anesthetized by i.p. injection of xylazine (12 mg/kg BW, Ursotamin^®^, Serumwerke Bernburg, Bernburg, Germany) and ketamine (90 mg/kg BW, Rompun^®^, Bayer, Leverkusen, Germany). To analyze fracture healing in aged and young adult CD-1 mice, a well-established stable closed femoral fracture model, including intramedullary fixation with the MouseScrew (AO Foundation, Research Implants System, Davos, Switzerland), was used (Figure 1a). Under aseptic conditions, a medial parapatellar incision was performed at the right knee and the patella was dislocated laterally. The marrow cavity was opened by drilling a hole (diameter of 0.5 mm) into the intercondylar notch. Subsequently, an injection needle with a diameter of 0.4 mm was placed into the intramedullary canal. Afterwards, a tungsten guidewire (Figure 1a; diameter of 0.2 mm) was inserted through the needle. After removal of the needle, the femur was fractured by a 3-point bending device (Figure 1b) and The MouseScrew was implanted over the guidewire to stabilize the fracture [14]. After fixation of the fracture, the wound was closed using 5-0 synthetic sutures. Adequate reduction of the fracture and position of the implant were confirmed by radiography (Figure 1c; MX-20, Faxitron X-ray Corporation, Wheelin, IL, USA). All fractures were simple, transverse midshaft fractures according to the AO classification type A2 fracture. Due to the minimally invasive approach and the high reliability of the present fracture model, no implant dislocation, infections or other complications were observed. Therefore, all animals survived until their designated time points of investigation.

### 2.3. Radiographic Analysis

Radiographic analysis was performed by using a radiographic score with stage 0 indicating radiological non-union, stage 1 indicating possible union and stage 2 indicating radiological union. The animals were re-anesthetized, and lateral X-rays (MX-20, Faxitron X-ray Corporation) of the healing femora were performed. To validate our results, the radiological scoring was performed by multiple observations. Moreover, the callus diameter/femoral diameter ratio (%) was measured to investigate the remodeling process during fracture healing.

### 2.4. Biomechanical Analysis

For biochemical analysis, femora were resected at 1, 2, 3, 4 and 5 weeks (*n* = 10 per study group) after fracture and freed from soft tissue. After removal of the implants, callus stiffness was measured using a three-point bending device (Mini-Zwick Z 2.5; Zwick, Ulm, Germany). To guarantee a standardized analysis, the ventral side of each femur was mounted downwards, and the callus was placed centrally between the stamps. The distance between the two stamps was 6 mm. The measurement was carried out at a constant speed of 1 mm/min, orthogonal to the axis of the femur. Due to the different time points of healing studied, the loads that had to be applied varied markedly between the individual animals. Loading was stopped individually in every case when the actual load–displacement curve deviated more than 1 % from linearity [16]. To account for differences in bone stiffness of the individual animals, the unfractured left femora were also analyzed, serving as internal control.

### 2.5. Histological Analysis

After biomechanical testing, bones were fixed in paraformaldehyde for 24 h. Subsequently, the specimens were embedded in a 30% sucrose solution for another 24 h and then frozen at −80 °C. Longitudinal sections through the femoral axis with a thickness of 4 mm were cut for histomorphometric analysis and stained with Safranin-O. At a magnification of 12.5× (Olympus BX60 Microscope, Olympus, Shinjuku, Japan; cellSens Dimension 1.11 software, Olympus), structural indices were calculated. The following histomorphometric parameters of the bone defects were evaluated: (i) ratio of cartilaginous tissue, (ii) ratio of fibrous tissue and (iii) ratio of bone tissue. Each area was marked and calculated using the ImageJ analysis system (NIH, Bethesda, MD).

### 2.6. Statistics

All data are given as means ± SEM. After proving the assumption for normal distribution (Kolmogorov–Smirnov test) and equal variance (F-test), comparisons between the two experimental groups were performed by a Student’s *t*-test. For non-parametrical data, a Mann–Whitney U test was used. Statistics were performed using SigmaPlot 13.0 software (Systat Software GmbH, Erkrath, Germany). A *p*-value < 0.05 was considered to indicate significant differences.

## 3. Results

### 3.1. Radiographic Analysis

In both study groups, the X-rays revealed typical patterns of secondary fracture healing associated with the formation of large fracture calluses and subsequent bone remodeling (Figure 2a–j). The analysis revealed a slightly increased radiographic score during the entire observation period in young adult mice when compared to aged animals (Figure 3a). However, these differences did not prove to be statistically significant (*p* ≥ 0.05).

Analysis of the remodeling process by evaluation of callus diameter to femoral diameter revealed a lower ratio at 2 weeks after fracture in aged animals (Figure 3b). At 3, 4 and 5 weeks after fracture, however, the analysis showed a higher ratio of callus diameter to femoral diameter in aged mice when compared to young adult animals (Figure 3b). Of note, at 1 week after fracture, the ratio of callus diameter/femoral diameter did not differ between the two study groups.

### 3.2. Biomechanical Analysis

Biomechanical analysis of the bending stiffness of fractured femora revealed a significantly decreased bending stiffness at 3 and 4 weeks after fracture in aged animals when compared to young adult mice (Figure 4a,b). This decreased bending stiffness indicated an impaired quality of newly formed bone as well as a deceleration of fracture healing in aged animals. Of interest, the significant differences were observed when comparing the absolute bending stiffness data (Figure 4a) as well as when comparing the relative (to nonfractured contralateral bone) bending stiffness data (Figure 4b). In contrast, at 1, 2 and 5 weeks after fracture, we observed no significant differences in the absolute and relative bending stiffness data of the fractured femora between the two study groups (Figure 4a,b).

### 3.3. Histological Analysis

The histological analysis demonstrated clear signs of endochondral fracture healing in young adult and aged mice. The quantitative analysis showed a significantly lower ratio of cartilaginous tissue at 1 week after surgery in aged mice (Figure 5a,b and Figure 6a). At 2 and 3 weeks, however, the amount of cartilaginous tissue was significantly higher when compared to young adult mice (Figure 5c–f and Figure 6a). This may indicate a delayedendochondral ossification in aged animals. Notably, at 4 and 5 weeks after surgery, only remnants of cartilaginous tissue were observed in both study groups (Figure 5g–j and Figure 6a). The ratio of fibrous tissue was significantly increased at 1 and 2 weeks after surgery in aged animals when compared to that in young adults. At 3, 4 and 5 weeks, however, the ratio of fibrous tissue did not differ between aged and young adult mice (Figure 5a–j and Figure 6b). Further analysis demonstrated a lower ratio of bone tissue in aged animals at 2 and 3 weeks after surgery. At 1, 4 and 5 weeks, the analysis showed no significant differences between the two study groups (Figure 5a–j and Figure 6c).

## 4. Discussion

Fracture models in mice represent a powerful tool to investigate radiographic and biomechanical healing properties during bone regeneration. With an increasing elderly population, the treatment of geriatric patients becomes of vital importance in trauma and orthopedic surgery. However, the impact of aging on the mechanical quality of the fracture callus and newly formed bone tissue is still controversially discussed and remains largely unclear. Accordingly, we analyzed in the present study the femoral fracture healing in young adult and aged CD-1 mice by X-rays, biomechanics and histology. Our results indicate a deceleration of callus remodeling and fracture healing in aged animals as demonstrated by an increased callus diameter to femoral diameter ratio, a delay in endochondral ossification and a decreased bending stiffness at 3 and 4 weeks after surgery when compared to young adult mice. However, the overall healing capacity in aged animals was not impaired, thus disproving our initial hypothesis.

During the last decades, murine fracture healing models have become of increasing interest in preclinical trauma research. Low housing costs and shorter breeding cycles present a distinct advantage when compared to larger animals. Moreover, when using mice, there is a large array of gene-targeted strains and antibodies available to specifically target a molecular pathway of interest. However, the small size of rodents, especially of mice, represents a major challenge in the development of sufficient fracture healing models [5]. Accordingly, in some experimental studies, the fractured bone has been left unstabilized [17,18]. This approach, however, displays major disadvantages regarding animal welfare, such as pain management and alteration of mobility. Furthermore, in unstabilized fracture models, the process of bone healing is analyzed under non-standardized conditions. Therefore, murine fracture healing models should use implants, which provide both axial and rotational stability. A wide range of preclinical fracture models have recently been developed. These include intramedullary pins, locking nails, pin-clip devices, locking plates, external fixators, and intramedullary compression screws [5].

In rodents, the femur and tibia are appropriate for the study of fracture healing. The tibia only possesses thin tissue coverage, thus, a closed fracture model can be easily performed. However, some details have to be considered when using the tibia for fracture healing analysis: (i) a potential fracture of the fibula may affect the stability of the tibia fixation, (ii) the accuracy of the biomechanical analysis of bone healing may be affected by the tapered shape of the bone, (iii) due to the thin soft tissue coverage, the tibia should not be preferred when analyzing the role of soft tissue on bone repair. The femur, on the other hand, is a long tubular bone with a consistent and larger diameter, when compared to the tibia. Hence, the femur allows the application of sophisticated implants such as screws and plates. Accordingly, the femur may be preferred when analyzing long bone fracture healing when good muscle coverage is required, or potential effects of soft tissue injury are investigated [5]. In the present study, we used a closed femoral fracture model with an intramedullary compression screw (MouseScrew) [14]. The screw is made of a distal cone-shaped head and a proximal thread allowing interfragmentary compression, thus achieving both rotational and axial stability. This fracture model guarantees a minimally invasive approach and high standardization due to the simple surgical technique.

Aging leads to severe physiological changes in the skeletal system, particularly in regard to angiogenesis and inflammation. Successful bone regeneration requires adequate angiogenesis and vascularization of the callus tissue. The vasculature provides the fracture zone with vital nutrients and cells, including granulocytes, macrophages, and stem cells. These cells recruit osteogenic and chondrogenic progenitor cells to the fracture site, which form stable callus tissue at the fracture site. Subsequently, a remodeling cascade is initiated, leading to the removal of excessive bone tissue and ultimately to the formation of mature lamellar bone. During this cascade, a plethora of mediators are involved, including BMP-2 and BMP-4, receptor activator of NF-κB ligand (RANKL), a stimulator of osteoclastogenesis, osteoprotegerin (OPG), an inhibitor of osteoclastogenesis, as well as vascular endothelial growth factor (VEGF). Interestingly, VEGF is not only a stimulator of angiogenesis, but is also directly involved in osteoclast recruitment and differentiation, thus directly contributing to the process of endochondral ossification [19]. A disturbance of the angiogenic response, however, can lead to impaired or even failed fracture healing. Several experimental studies demonstrated that the pharmacological blockade of angiogenesis by TNP-470 or non-steroidal anti-inflammatory drugs (NSAIDs) hampers bone regeneration and may even lead to non-union formation [20,21].

In aged individuals, angiogenesis at the fracture is hampered by an age-related dysfunction of the bone vascular system. The vascular perfusion of the skeleton deteriorates, as demonstrated by decreased patency of bone marrow blood vessels in older rats compared to young [22]. The reduced vascularization at the fracture site in aged individuals may also delay angiogenesis during fracture healing, as indicated by a reduced amount of blood vessels in aged mice when compared to young adults at early healing time points [22]. This is associated with reduced expression of VEGF, hypoxia inducible factor 1α (HIF-1α) and matrix metallopeptidase (MMP) 9 and 13 within the callus tissue of aged mice when compared to young [22]. Moreover, aging is associated with a reduction of pericytes within the skeletal system [19]. Pericytes are not only essential for the stabilization and development of microvessels, but also posses the ability to differentiate to osteoblasts and osteoclasts, therefore, directly contributing to fracture healing. Hence, it may be speculated that the age-related loss of pericytes within the bone vascular system directly contributes to decelerated and impaired angiogenesis and bone regeneration in the aged [19].

The inflammatory response after fracture is the first stimulus to initiate the process of fracture repair. Neutrophil granulocytes arrive at the fracture site and recruit macrophages and other inflammatory cells. The cytokines secreted by the macrophages attract mesenchymal stem and progenitor cells in order to form the callus tissue and start the repair process [23]. However, in addition to an acute inflammatory response, the resolution of acute inflammation is also essential for successful bone regeneration. Interestingly, aged individuals demonstrate a low-grade, chronic inflammation despite the absence of acute infection [24]. This dysregulated, chronic state of inflammation in geriatric patients may hamper the proper initiation of fracture repair.

During the acute inflammatory response, M1 macrophages are activated. The resolution of the inflammatory phase is characterized by the repolarization of macrophages to an alternatively activated M2 phenotype, which is vital for tissue regeneration [25]. Notably, the chronic inflammation seen in aged individuals leads to a failure of repolarization of macrophages from the M1 to M2 phenotype, resulting in increased bone resorption and decreased bone formation [25]. Moreover, osteoclastogenesis and bone resorption are stimulated by persisting inflammatory cytokines such as TNF-alpha during aging [26]. In addition, gene expression analysis of fracture calluses 2 weeks after injury in young adult and aged animals demonstrated an upregulation of several toll-like receptors (TLRs) in aged animals. Notably, the inhibition of MyD88, an adapter protein that controls TLR signaling by a small molecule inhibitor, resulted in an improved process of bone healing indicated by novel bone and callus formation [27]. These findings indicate that the modulation of the inflammatory response by targeting specific pro-inflammatory cytokines may be a promising therapeutic approach in clinical practice to improve fracture healing in geriatric patients.

The interaction between mesenchymal stem and progenitor cells at the fracture site is of vital importance for successful bone regeneration. This crosstalk may be impaired in tissues with age-related chronic inflammation. In fact, these tissues are characterized by the secretion of a senescence-associated secretory phenotype (SASP), which directly stimulates cell senescence [28]. Recent evidence demonstrates the presence of SASP in the skeletal environment, resulting in cellular senescence and impaired function of stem cells [29]. Inflammatory mediators that contribute to this effect include TNF-alpha, interleukin (IL)1 and nuclear factor kappa-light-chain-enhancer of activated B cells (NF-kappaB) [30]. Conversely, inhibiting these inflammatory mediators and specifically targeting stem cells at the fracture site may be a promising therapeutic approach to overcome the detrimental effects of aging on fracture healing.

The age-related effects on angiogenesis and inflammation may significantly alter the quality of fracture repair in aged individuals. In fact, frequently observed failures of implant fixation in aged patients indicate a reduced healing potential [31] and an increased rate of delayed healing and non-unions. Another age-related disorder that may contribute to an impaired ability to regenerate bone in the aged is osteoporosis. Namkung et al. [32] demonstrated a significant reduction in callus size, bone mineral density and mechanical strength in ovariectomy-induced osteoporotic rats at 3 weeks after fracture. Additionally, Wang et al. [33] found a lower callus bone mineral density, callus failure stress as well as a delay in bone formation in osteoporotic rats. This was associated with loosely and irregularly arranged novel bone trabeculae. On the other hand, several other authors reported that osteoporosis does not affect the fracture healing process. Langeland et al. [34] showed that the tensile strength and collagen content of fractured tibial bone does not differ between ovariectomized rats and controls. These findings are supported by an experimental study of Melhus et al. [35] demonstrating that osteoporosis induced ovariectomy and vitamin D deficiency have no significant negative effects on bone healing in rats.

In the present study, we found a significantly increased callus diameter to femoral diameter ratio in aged animals at later time points after fracture, indicating a delay in callus remodeling. After fracture and the development of a soft callus, this temporary structure is gradually remodeled from calcified cartilage tissue to novel bone [36]. Apparently, this process is delayed in aged CD-1 mice when compared to young adult animals. Appropriate callus remodeling is a crucial prerequisite for successful bone regeneration as well as biomechanical strength and resilience. Accordingly, we also found a significantly decreased bending stiffness at 3 and 4 weeks after fracture in aged mice when compared to young adult animals. Notably, this significant difference was observed when evaluating the absolute data as well as the relative data referring to the unfractured contralateral femora. Surprisingly, we found no significant difference in bending stiffness at 5 weeks after fracture between the two study groups. Apparently, although the process of fracture repair is delayed in aged CD-1 mice, the overall healing capacity of the femora is not significantly affected by aging. Accordingly, our histological analysis demonstrated a delayed endochondral ossification in aged animals as well as a higher amounts of fibrous tissue at early healing time points. The amount of bone tissue, however, did not differ at 4 and 5 weeks after surgery between the two study groups. These findings are in line with our previous study investigating the effects of osteoporosis in senescence-accelerated SAMP6 mice [37]. Herein, the biomechanical analysis showed a decrease in bending stiffness at 2 weeks after fracture, whereas at 5 weeks, no significant difference was observed in 10-month-old SAMP6 mice when compared to controls. Lopas et al. [38] analyzed the callus expansion and bone volume in young adult (5-month-old) and geriatric (25-month-old) mice by µCT-analysis. The authors demonstrated that aged animals possessed smaller fracture calluses and a delayed healing progress. However, the capacity for endochondral and intramembranous ossification was still intact in aged mice. In addition, the authors found no significant differences in bone volume fraction at later healing time points between young adult and aged animals. Therefore, these findings are in line with our results, as our histological analysis demonstrated no significant differences in callus composition and biomechanical stiffness between the two study groups at 4 and 5 weeks after surgery.

Even though osteoporosis has a major impact on bone quality and metabolism in aged individuals, there are a plentitude of other factors and comorbidities that have a potential impact on fracture healing, such as peripheral arterial occlusive disease, diabetes mellitus or age-related changes in the inflammatory response [39]. These factors are left out in artificial aging models, such as ovariectomy-induced osteoporosis or transgenic senescence-accelerated mouse strains. To overcome this problem, we used CD-1 mice, a wildtype outbred mouse strain, to mimic the aging process of patients in clinical practice as accurately as possible. In addition, we used a sophisticated fracture model with an intramedullary screw [14], providing both axial and rotational stability. In most previous studies [32,38], a single intramedullary pin was used for fracture stabilization. However, we feel that this surgical approach lacks the necessary rotational stability required to mimic an osteosynthesis performed in clinical practice. Moreover, most studies investigated the effect of aging in C57BL/6 mice [38]. Notably, there is evidence that the bone mineral maturity and density as well as the biomechanical properties of bone show significant differences between C57BL/6 and CD-1 mice [40,41] Accordingly, the process of fracture repair also may differ between these two mouse strains. To our knowledge, the present study is the first to analyze the effects on fracture repair in CD-1 wildtype mice by radiological, biomechanical as well as histological analysis.

One potential limitation of the present study was the fact that the sex of the animals may have an influence on bone regeneration and the resulting bending stiffness of the mice femora. However, due to the long breeding time of the aged mice, we used both male and female animals to acquire the necessary number of animals for the analysis.

In conclusion, we demonstrated that aged CD-1 mice show a delayed callus formation and remodeling process, leading to a reduced bending stiffness at 3 and 4 weeks after fracture. However, we found no significant difference in the biomechanical properties and callus composition of aged animals at 5 weeks after fracture when compared to young adult CD-1 mice, indicating that the overall healing capacity of the fractured femora is not affected by aging.

## Figures and Tables

**Figure 1 bioengineering-10-00275-f001:**
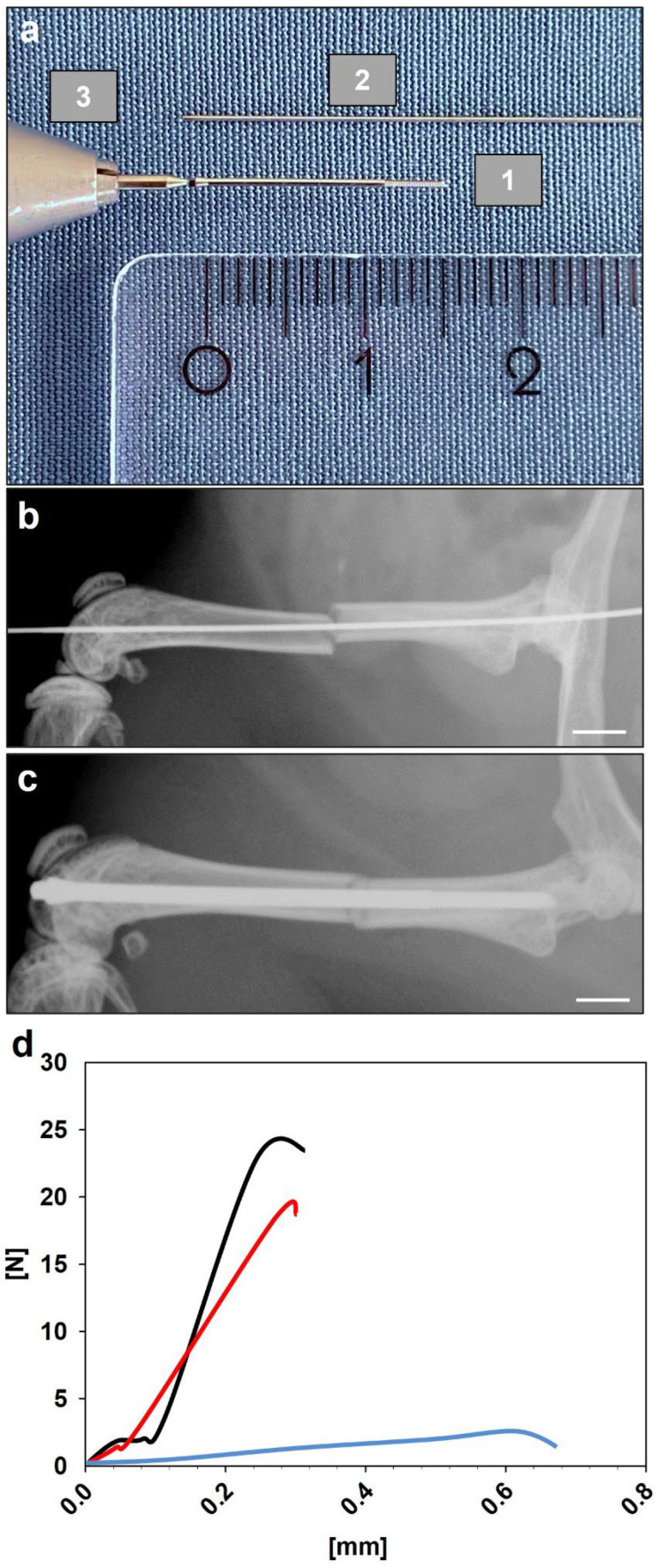
(**a**) Illustration of the intramedullary MouseScrew (1), the tungsten guide-wire (2) and the screw holder (3) used in the present fracture model. Scale bar: 2 mm. (**b**) Fractured femora with inserted tungsten guidewire to guide the MouseScrew during insertion. Scale bar: 2 mm. (**c**) Fractured femora after intramedullary stabilization with the MouseScrew. Scale bar: 2 mm. (**d**) Representative load displacement curves of an intact femur (black), 1 week after fracture (blue) and 5 weeks after fracture (red) in young adult animals.

**Figure 2 bioengineering-10-00275-f002:**
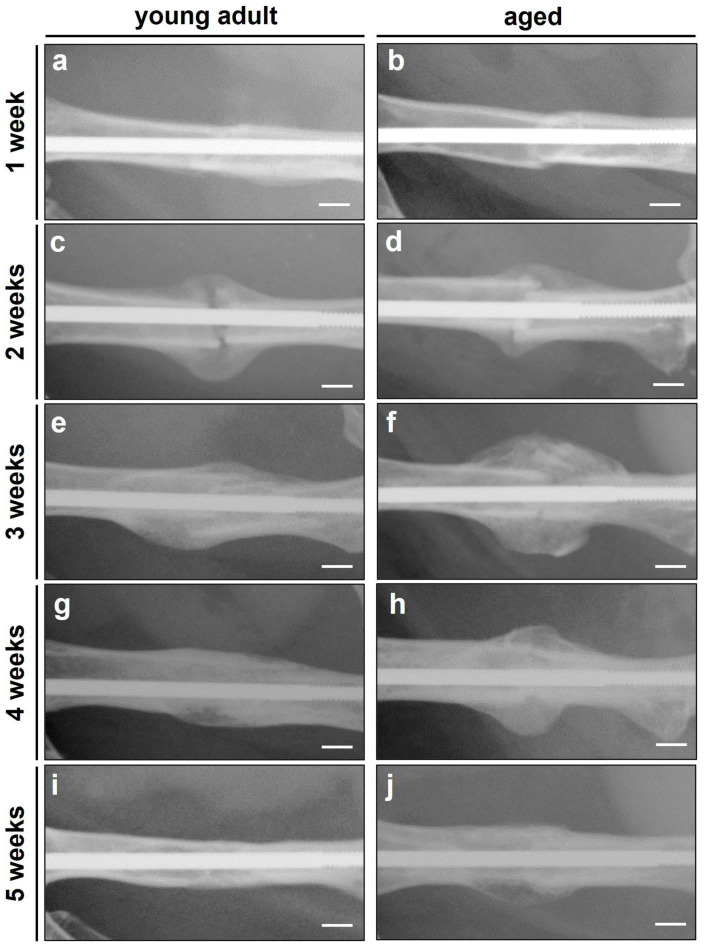
(**a**–**j**) Representative X-rays of fractured femora at 1 (**a**,**b**), 2 (**c**,**d**), 3 (**e**,**f**), 4 (**g**,**h**) and 5 weeks (**i**,**j**) after fracture in young adult (**a**,**c**,**e**,**g**,**i**) and aged (**b**,**d**,**f**,**h**,**j**) CD-1 mice. Scale bars: 1 mm.

**Figure 3 bioengineering-10-00275-f003:**
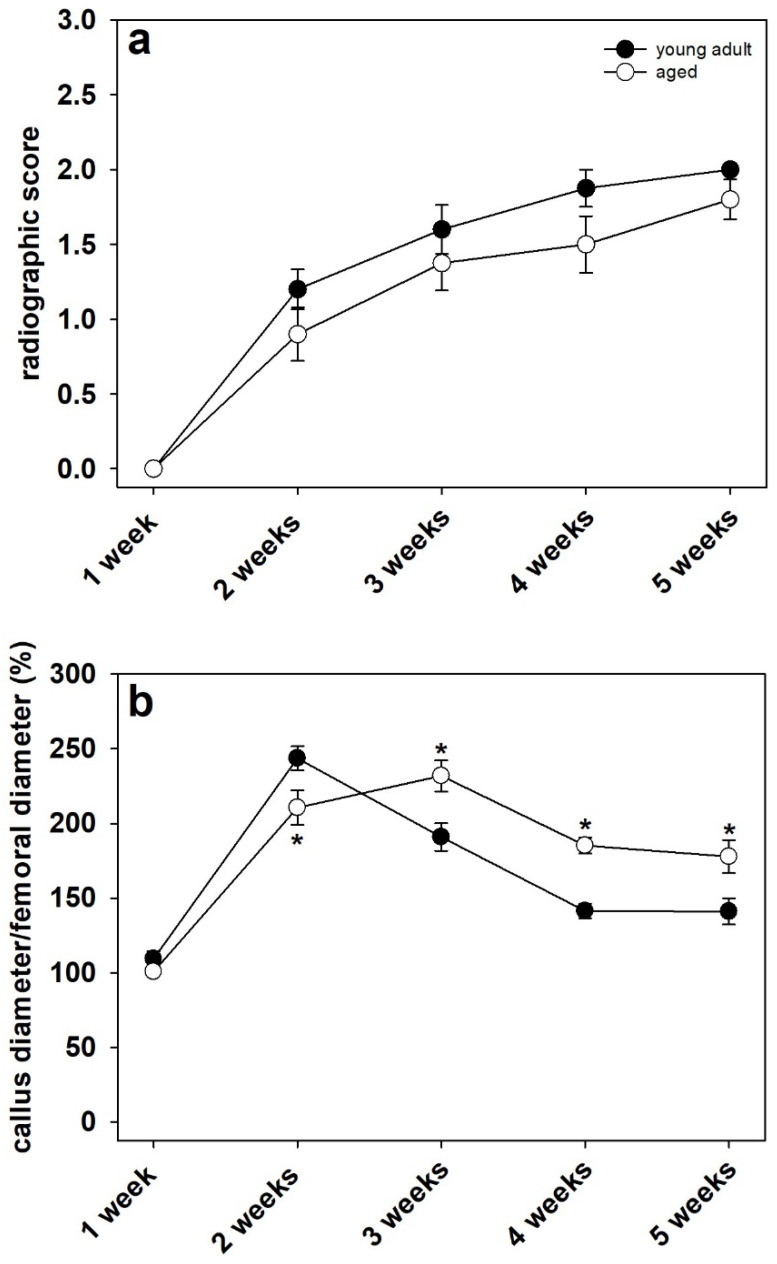
(**a**) Analysis of the radiographic score at 1, 2, 3, 4 and 5 weeks after fracture in young adult (black circles, *n* = 10) and aged (white circles, *n* = 10) CD-1 mice (Mann–Whitney U test used at 1, 2, 3, 4 and 5 weeks). (**b**) Analysis of the callus diameter/femoral diameter (%) at 1, 2, 3, 4 and 5 weeks after fracture in young adult (black circles, *n* = 10) and aged (white circles, *n* = 10) CD-1 mice (Mann–Whitney U test used at 1 week, Student’s *t*-test used at 2, 3, 4 and 5 weeks). Means ± SEM; * *p* < 0.05 vs. young adult.

**Figure 4 bioengineering-10-00275-f004:**
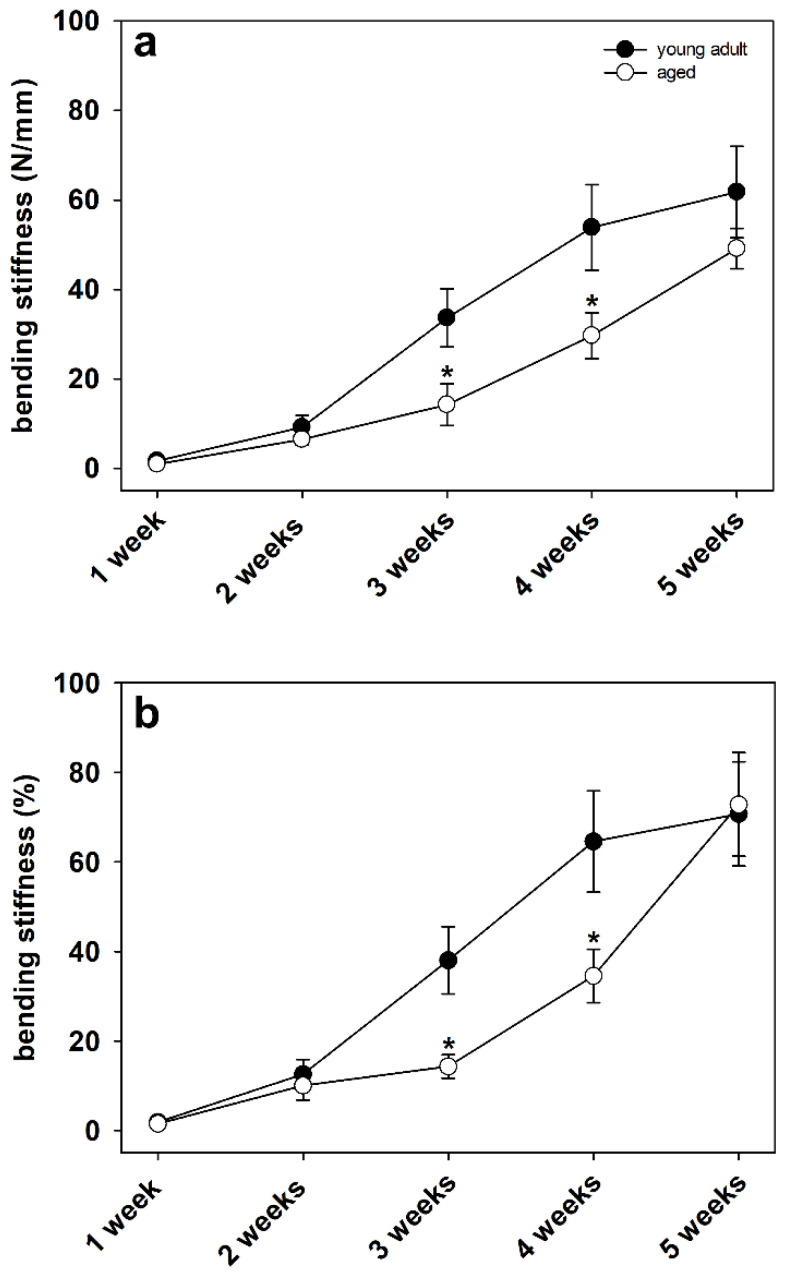
(**a**) Analysis of the bending stiffness (N/mm) of the fractured femora at 1, 2, 3, 4 and 5 weeks after fracture in young adult (black circles, *n* = 10) and aged (white circles, *n* = 10) CD-1 mice (Mann–Whitney U test used at 1, 2 and 5 weeks, Student’s *t*-test used at 3 and 4 weeks). (**b**) Analysis of the bending stiffness (%) of fractured femora in relation to the corresponding unfractured contralateral femora at 1, 2, 3, 4 and 5 weeks after fracture in young adult (black circles, *n* = 10) and aged (white circles, *n* = 10) CD-1 mice (Mann–Whitney U test used at 1, 2, 3 and 5 weeks, Student’s *t*-test used at 4 weeks). Means ± SEM; * *p* < 0.05 vs. young adult.

**Figure 5 bioengineering-10-00275-f005:**
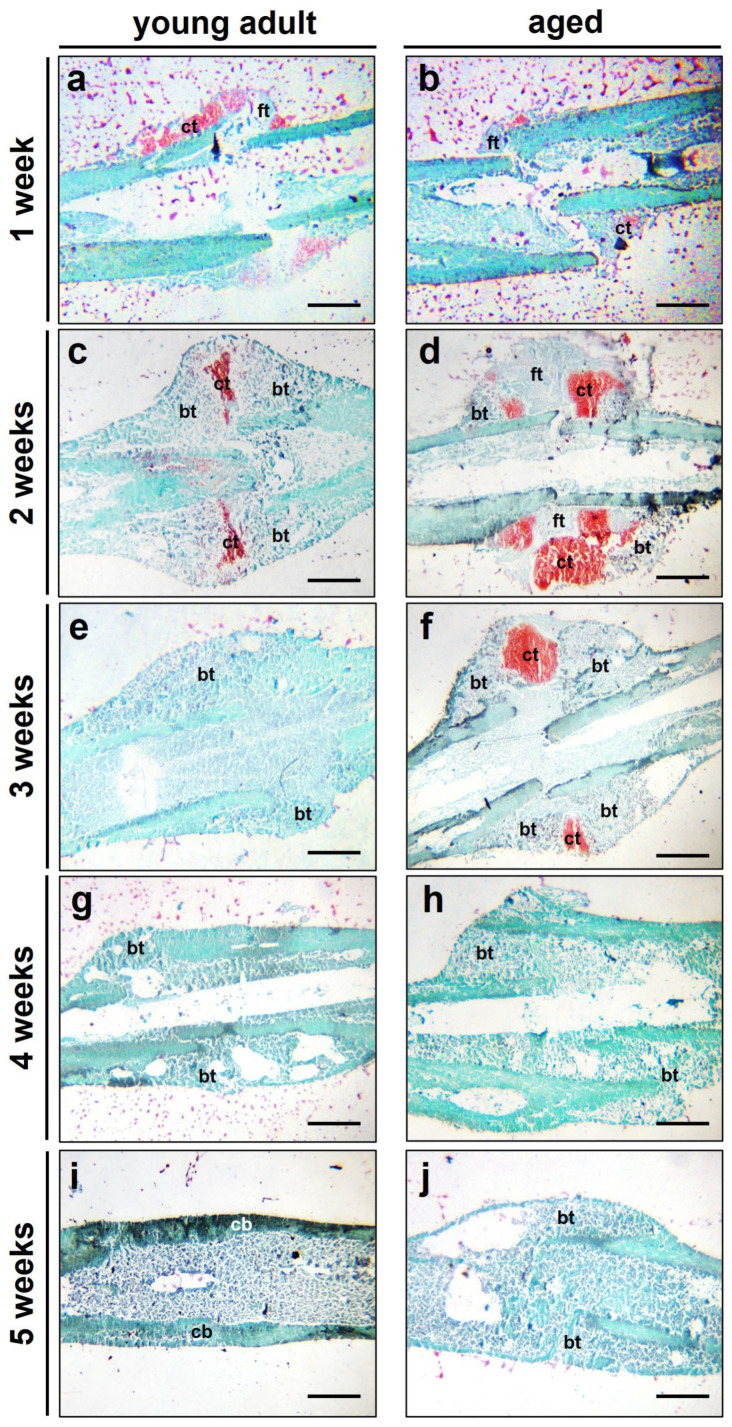
(**a**–**j**) Representative histological images (Safranin-O-staining) of fractured femora at 1 (**a**,**b**), 2 (**c**,**d**), 3 (**e**,**f**), 4 (**g**,**h**) and 5 weeks (**i**,**j**) after fracture in young adult (**a**,**c**,**e**,**g**,**i**) and aged (**b**,**d**,**f**,**h**,**j**) CD-1 mice (ct: cartilaginous tissue, ft: fibrous tissue, bt: bone tissue, cb: cortical bone). Scale bars: 1 mm.

**Figure 6 bioengineering-10-00275-f006:**
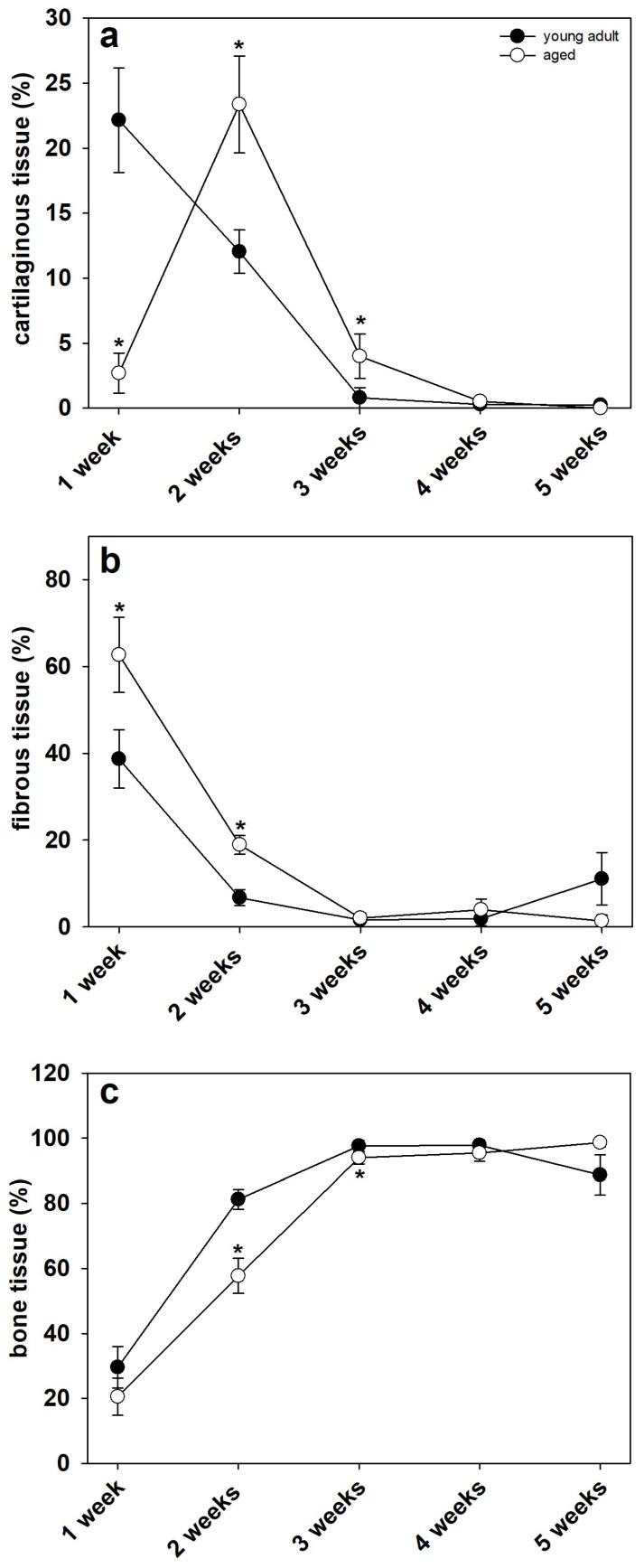
(**a**) Analysis of the ratio of cartilaginous tissue within the calluses of the fractured femora at 1, 2, 3, 4 and 5 weeks after fracture in young adult (black circles, *n* = 10) and aged (white circles, *n* = 10) CD-1 mice (Mann–Whitney U test used at 1, 3, 4 and 5 weeks, Student’s *t*-test used at 2 weeks). (**b**) Analysis of the ratio of fibrous tissue within the calluses of fractured femora at 1, 2, 3, 4 and 5 weeks after fracture in young adult (black circles, *n* = 10) and aged (white circles, *n* = 10) CD-1 mice (Student’s *t*-test used at 1 week, Mann–Whitney U test used at 2, 3, 4 and 5 weeks). (**c**) Analysis of the ratio of bone tissue within the callus tissue of fractured femora at 1, 2, 3, 4 and 5 weeks after fracture in young adult (black circles, *n* = 10) and aged (white circles, *n* = 10) CD-1 mice (Student’s *t*-test used at 1 and 2 weeks, Mann–Whitney U test used at 3, 4 and 5 weeks). Means ± SEM; * *p* < 0.05 vs. young adult.

## Data Availability

The data presented in this study are available on request from the corresponding author.

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
