# Peer review of "Radiographic, Biomechanical and Histological Characterization of Femoral Fracture Healing in Aged CD-1 Mice"

_bioengineering, 2023, doi:10.3390/bioengineering10020275_

Round 1
Reviewer 1 Report
Bioengineering Manuscript Review
Manuscript Title: Radiographic and biomechanical analysis of femoral fracture healing in senescent CD-1 mice
Manuscript Type: Full length article
Authors: Menger M, Manuschewski R, Emmerich M, Tobias AL, Menger MD, Laschke MW, and Histing T.
Manuscript Summary
This study by Menger et al. compares the fracture healing potential of young adult and naturally aged mice using radiographic and biomechanical measures. The experiments suggest that aged mice have a slower healing response that eventually achieves a radiographically and biomechanically equivalent healing result compared to young mice.
Manuscript Strengths
The manuscript is clear and well-written, with a simple design appropriate to the research objective. There are no major flaws in the research design.
Manuscript Weaknesses
Main Concerns
There are only two major concerns with this manuscript that require substantial responses and revisions from the authors:
1. The choice of the three-point bending test may be problematic if the loading point in the middle of the span was placed at the site of the fracture/callus. Additionally, the data processing method is not clearly described. Representative load-displacement curves need to be provided for an early-stage (1 week) and late-stage (5 week) healing case as well as an intact case. Is the 1% criterion for the stiffness calculation based on data post-processing or was there an active control program to suspend the test? How was the linear region of the test curve defined? These features of the data need to be shown, annotated, and explained in the methods. Additionally, the dimensions of the three-point bend fixture and the direction of bending need to be reported.
2. The statistics are incorrect for time and between-groups effects. Multiple t-tests were used, one at each timepoint, to compare the young and aged mouse groups. A more correct approach would be a two-way ANOVA, with time and group as factors, followed by post-hoc testing at each timepoint if the overall effect of group is significant. Additionally, the methods report the use of both parametric and non-parametric tests, but the results do not state which test was used for which datasets. This needs to be clarified.
Minor Concerns
The following minor concerns should also be addressed through minor revisions:
No sample size justification or powering data was provided for the choice of number of animals. Was any such data available?
No animal complications or unplanned euthanasia events were reported. Did all animals survive to their designed final timepoints? Any excluded animals or protocol deviations need to be reported.
The use of the term “Goldberg classification” for the radiographic score is misleading and should be changed for two reasons. First, the “Goldberg classification” in orthopaedic trauma is a classification system for periprosthetic fractures associated with TKA. Second, the radiographic scoring system being used here is not the same as the one reported in the Goldberg 1985 paper. That scoring system was for bone grafts and was out of a total score of 7. The radiographic scores here are fine, but they should not be called “Goldberg” scores or classifications.
How do you know the aged mice were senescent? Was there any biomarker confirmation of senescence-associated secretory phenotype? If no biomarker validation of senescence is available for this experiment, then the term “aged” should be used exclusively instead of “senescent”.
The manuscript would be stronger with the addition of a stated hypothesis for the experiment.
It is slightly misleading to refer to the slower healing in aged mice as “impaired” because the end result of the healing process was equivalent to the young mice, according to the radiographic and biomechanical measurements used. Please consider replacing the word “impaired” with a more accurate adjective like “delayed” or “slower” throughout the manuscript.
Reviewer 2 Report
This manuscript seeks to determine the effects of aging on fracture healing in CD-1 mice. This study is well executed but limited by its cursory evaluation of the fracture callus.
Major concerns
The study group includes 100 male and 100 female mice. However, it appears that the data from each sex were pooled therefore obscuring any potential sex differences in response. Male and female fracture healing data should be presented separately.
There is no histological evaluation of the callus. Fracture healing is only evaluated radiographically and by mechanical testing. The group describe changes in callus diameter but without histology we do not know why this might be – this seems to be a significant missed opportunity.
There are a number of studies that have evaluated the effects of age on fracture healing in murine models, for example Lopas et al 2014 Fracture in geriatric mice show decreased callus expansion and bone formation Clin Ortho Relat res. The current data is not discussed in relation to existing data in this area.
Given existing data on aging and fracture healing in murine models it is not clear how this study fills a gap in knowledge. Fracture healing in CD-1 mice may not have been clearly described but how this model is superior to other murine models, and how this advances our understanding of healing in an aged population should be addressed
Reviewer 3 Report
This study investigated the effect of ageing on the radiographic and biomechanical properties during fracture healing. Here summarizes my comments:
1. The methodology set up in this study was simple and easy to follow which directly showed there was difference in callus remodelling rate between young and aged mice but the healing capacity would reach similar levels when time was allowed for the bone to heal. However, it would be better if there is more elaboration on the assessment methods and results.
2. The Goldberg scoring system had a narrow range of data variation and the reliability was rather inconclusive. Different individual assessors may be needed to confirm the scoring where guidelines or more descriptive information (radiopacity or fracture gap size) should be supplemented to ensure consistency in classifying the healing status.
3. The use of CD-1 mice for research in ageing and fracture was well supported by previous studies but the novelty of using this model rather than C57BL6 was not clearly discussed. There could be more information on the difference between CD-1 and C57 mice models, such as their genetic background and immuno-inflammatory response in fracture repair.
4. Estrogen deficiency in aged female mice would lead to osteoporosis and had proven to be playing a role in fracture healing mechanism. The difference in BMD and bone microarchitecture between male and female mice would result in large standard deviation when comparing their mechanical properties and healing status. As the mice were only randomised into young and aged groups, the variation within group may also be considered in deciding the sample size. Please comment.
Round 2
Reviewer 2 Report
I believe combining male and female cohorts without testing that there are no major differences remains a serious flaw in the approach.
